# Tannins-Based Extracts: Effects on Gut Chicken Spontaneous Contractility

**DOI:** 10.3390/molecules28010395

**Published:** 2023-01-03

**Authors:** Laura Beatrice Mattioli, Ivan Corazza, Matteo Micucci, Marco Pallavicini, Roberta Budriesi

**Affiliations:** 1Food Chemistry and Nutraceutical Lab, Department of Pharmacy and Biotechnology, Alma Mater Studiorum-University of Bologna, 40126 Bologna, Italy; 2Department of Specialistic, Diagnostic and Experimental Medicine (DIMES), University of Bologna, S. Orsola-Malpighi University Hospital, Alma Mater Studiorum-University of Bologna, 40126 Bologna, Italy; 3Department of Biomolecular Sciences, University of Urbino “Carlo Bo”, 61029 Urbino, Italy; 4Department of Pharmaceutical Sciences, University of Milano, 20129 Milan, Italy

**Keywords:** tannins, chestnut: Silvafeed ENC, *Castanea sativa* Mill. wood, Quebracho: Silvafeed Q, *Schinopsis balansae* Engl. hardwood, broiler chickens, ileum, proximal colon, gallbladder, caecum, spontaneous contractility

## Abstract

The impossibility of using drugs for the health of farm animals leads to the search for alternative strategies with two purposes: to maintain animal health and safeguard human health. In this perspective, tannins have shown great promises. These phytocomplexes obtained from natural matrices with multiple health properties may be used as a feed supplement in chicken farms. In this work, we studied two tannin-based extracts (from *Castanea sativa* Mill. wood and from *Schinopsis balansae* Engl. Quebracho Colorado hardwood) with different chemical compositions on the spontaneous contractility on the isolated intestinal tissues of healthy chicken. The results showed that the chemical composition of the two phytocomplexes influenced the spontaneous intestinal contractility in different ways by regulating the tone and consequent progression of the food bolus. The chemical analysis of the two extracts revealed that *Castanea sativa* Mill. wood mainly contains hydrolysable tannins, while *Schinopsis balansae* Engl. hardwood mainly contains condensed tannins. The two phytocomplexes showed different effects towards gastrointestinal smooth muscle contractility, with *Castanea sativa* Mill. wood providing a better activity profile than *Schinopsis balansae* Engl. hardwood.

## 1. Introduction

In intensive farming, the animals destined for human consumption are exposed to numerous diseases of bacterial and viral origin that affect the gastrointestinal tract. Currently, to treat infections, feed-additive antibiotics have provided beneficial responses in the food-producing animal and poultry industries, but antibiotic residues can be found in animal-derived products [1].

Therefore, the identification of nutraceuticals endowed with antimicrobial activities and with other effects, such as anti-inflammatory, antioxidant and gut modulatory properties, may lead to an improvement in the general health status of animals and to a decrease in their antibiotic needs, which may result in a reduction of antimicrobial resistance [2].

The transmission of disease is favored by liquid excrements, which, by stagnating in the litter, greatly facilitate the passage of bacteria and viruses through lesions of the legs, which often occur in breeding chickens. Improving both the quality of life and the welfare of farmed animals are objectives that benefit not only the producer, who provides a better quality of meat by lowering production costs, but also the consumer. In fact, the preventive use of antibiotics in animals raised for food is a major contributor to antibiotic resistance, which is a global threat to public health as people can develop sill from foodborne infections or from contact with animals and their environment [3].

This is particularly important in chicken farms, considering the fact that new legislation has been approved by The European Parliament to ban the prophylactic use of antibiotics, coming into force in 2022 [4]. Breeders are therefore looking for natural bioactive substances that, when added to the diet, improve the quality of life of the animals themselves without compromising the quality of the meat.

Supplements used for this purpose include tannins obtained from *Castanea sativa* Mill., a tree that grows in Europe and is renowned for its fruits and chestnuts and is rich in mineral salts and proteins, and tannins obtained from *Schinopsis balansae* Engl. Hardwood.

In addition to the important nutritional properties of the fruit, the heartwood represents an important source of bioactive molecules, such as tannins, endowed with several biological effects, of which the main ones are antiviral and antibacterial, astringent and antispasmodic and spasmolytic properties. The biological activity of wood extract, already documented in folk medicine, has been widely confirmed through studies that have demonstrated its efficacy in integrated therapy for human diseases. In particular, recent studies have confirmed the antidiarrheal action that occurs through an integrated mechanism that involves different receptors and channels [5]. This extract has no effect on the biliary system even in the conditions of a fat diet [6]. Other activities of this phytocomplex that may result in a quality improvement of an animal’s health status are the anti-inflammatory and antioxidant effects. In particular, this phytocomplex was shown to exert cardiovascular protective effects in healthy and obese animals [7,8,9].

Quebracho extract was used as it contains high amounts of condensed tannins and is endowed with many biologically beneficial effects. In addition, it was chosen in order to make a comparison between condensed and hydrolysable tannins in order to understand which molecules may be further studied for applications in animal farming.

The control of contractility in birds is quite similar to that of mammalians [10], thus suggesting that the action of natural extracts on the intestinal tracts of chickens and humans would be similar [11]. To evaluate the effects on contractility, it is necessary to consider both induced and spontaneous motility: the first one depends on the effects of the extracts on the receptors and channels involved in contractility control; the spontaneous one is due to the direct effects of the extracts on the smooth muscle.

These assumptions fit perfectly with the need of livestock breeders, who are highly exposed to infectious risk, in order to find natural alternatives to antibiotics that do not impair the quality levels of meat placed on the market.

To achieve these objectives, the use of hydrolysable tannins from *Castanea sativa* Mill. and condensed tannins from *Schinopsis balansae* (Quebracho Colorado tree) as a feed additive aimed at increasing the consistency of the stools in order to limit the spread of microorganisms and indirectly reduce the incidence of pathologies. This is supported by interesting results documented in different scientific works [12] and by direct applications in farming [13]. Moreover, these tannin-rich extracts are chemically characterized and can be a valid starting point for identifying new scaffolds to assist in the treatment of human diseases and to improve the health of livestock.

The aim of this work was to study the in vitro effects of a hydrolysable tannin-rich extract from *Castanea sativa* Mill. (Silvafeed ENC) and a condensed tannin-rich extract from *Schinopsis balansae* Engl. (Silvafeed Q) on the intestinal tracts and the gallbladder basal spontaneous contractility of healthy chickens in order to investigate their possible application as a food supplement for industrial farms.

## 2. Results

### 2.1. Chemistry

The tannin-based extracts contain hydrolyzable tannins from *Castanea sativa* Mill. (Silvafeed ENC) and condensed tannins *Schinoipsis balansae* Engl. (Silvafeed Q), respectively, and were supplied by SilvaTeam, San Michele di Mondovì, Italy. The tannin percentage and other components are shown in Table 1.

The composition of Silvafeed ENC (ENC) has been clarified by MALDI-TOF mass spectrometry, which states that it contains a high concentration of ellagic-type hydrolysable tannins [14]. Tannins have been studied by this analytical technique, showing the molecular structure of many hydrolysable tannins, such as the c-glycosidic tannins vescalagin and castalagin [15]. Furthermore, pentagalloyl glucose can exemplify the chemistry of this tannin family (Figure 1).

The molecular composition of condensed tannins has been confirmed by MALDI-TOF mass spectrometry. In particular, Silvafeed Q (Q) is confirmed to be composed mainly by polyflavonoids of the type of profisetinidins [16]. More details have been successively reported [17], highlighting that Q is a mixture of proanthocyanidins, which are oligomers consisting of linear structures that could be exemplified by the tetramer represented in Figure 2.

Table 2 reports the fine composition in tannins and polyphenolic compounds concentration of ENC, and Q.

### 2.2. Spontaneous Contractility

All extracts containing tannins have been studied to verify their effects on the spontaneous contractility of certain intestinal tracts: the duodenum, caecum, ileum, colon and gallbladder. Cumulative concentration–response curves were constructed. For all traits, an example of spontaneous contractility (SC), amplitude of contraction and relative standard deviation (MCA ± SCV), % variation in tone (BSMA) and harmonic content of the signal Power Spectral Density (PSD) were reported for each extract. The information relating to the wave panel allowed us to make some considerations relating to the individual extracts on each intestinal tract, and to correlate a link between their activity and the chemical composition.

#### 2.2.1. Duodenum

There is a decrease in spontaneous contractility (Figure 3): Q has much greater action than ENC. More particularly, ENC reduces (10%) the tone already at 0.5 mg/mL and then stabilizes; the most potent is the Q, whose relaxing effect reaches around 40% at 10 mg/mL. As for PSD, a slight increase in low-frequency variability with ENC is observed. A relevant increase with Q is observed already from 0.1 mg/mL but with a new reduction to 5 mg/mL. However, the variability always remains greater than in basal conditions. This suggests that ENC relaxes the muscles without significantly influencing the peristalsis waves; on the contrary, Q clearly reduces them. An increase in transit speed due to Q can negatively affect the effectiveness of duodenum function.

#### 2.2.2. Caecum

Both extracts reduce the tone (Figure 4). In particular, at 1 mg/mL ENC relaxes spontaneous contractions by about 9%. Q stands in between the two, with the greater effect at 0.5 mg/mL, by 20%. As for the harmonic content of the signal (PSD), there are no significant changes in variability at any frequency. Only noteworthy effect is an increase in the low frequencies with Q at 5 mg/mL. ENC reduces peristalsis frequencies blandly and in a concentration-dependent manner, while Q has an up-and-down pattern that is less easy to control with concentration.

#### 2.2.3. Ileum

ENC little affected spontaneous contraction (SC) and Q decreased it, but at the highest concentration (Figure 5). Q is the most potent, relaxing by about 40% at 1 mg/mL; ENC relaxes by about 10% at the same concentration. Q is effective at a low concentration (1 mg/mL) up to 37%, but then the relaxation is not maintained. As for the wave panel, ENC and Q do not relevantly modify it.

#### 2.2.4. Colon

Q relaxes the colon, with a maximum effect at 1 mg/mL (tone reduction by 38%), then the effect fades (Figure 6). On the contrary, ENC induces a little contraction of up to 5 mg/mL. Q reduces the spontaneous variability at all frequencies of interest, from 0.5 mg/mL. ENC has no effects.

#### 2.2.5. Gallbladder

ENC reduces the tone. Q does not significantly modify the contractility of the gallbladder (Figure 7). Concerning the wave panels, both extracts produce an increase in contractility. The most potent is Q, where spontaneous low-frequency contractility increases at a concentration of 0.1 mg/mL. ENC also boosts the low frequencies. Both ENC and Q increase spontaneous contractility, but ENC in a concentration-dependent manner.

## 3. Discussion

Medicinal chemistry is always looking for new molecular scaffolds for the design of new drug classes. Nutraceuticals aim to modulate biological mechanisms through action on the same targets as drugs. The biological actions of phytocomplexes can also be extended to the feed supplementation for livestock animals. In fact, the legal obligation to ban antibiotics as growth promoters in poultry diets has prompted the search for alternative strategies to increase both animal welfare and the quality of meat.

For this reason, many plants and their extracts are being studied for application in supplementation because of their potential positive effects [18,19]. Among these, the hydrolyzable tannins of *Castanea sativa* Mill. were shown to exert positive effects on farm animals, since they improve the quality of meat [20,21], reduce the cholesterol content in eggs and improve the map of the lipid composition [22]. The addition of 0.2% of ENC to the chicken diet improves growth performance [23].

Both ENC and Q also have a positive impact on human health. This extract contains condensed tannins that are endowed with hypoglycemic, antioxidant [24] and bactericidal actions as demonstrated by in vitro studies [25]. In the same way, Q has an antimicrobial action on some avian viruses [26] and on *salmonellae* [27]. In particular, ENC and Q are probably the most promising tools to replace antibiotics as chicken growth promoters [28]. In addition, ENC and Q modulate the microbiota of chickens [29]. An interesting prebiotic effect is also described for humans [30]. It is well known that some foods contain substances that limit the absorption of nutrients and are considered anti-nutrients. However, some of these exert a beneficial action on the microbiota. The best known are the phytates, which are transformed into inositol by intestinal bacteria. Inositol exerts significant actions in the intestine health promoting actions in the intestine [31]. Positive effects have also been shown for some tannins.

In humans, hydrolyzable tannin (ENC) actions on intestinal motility are known; when added to other well-known antivirals and antibacterials, they are proposed for the control of diarrhea. In the same way, ENC gastrointestinal and antimicrobial effects may be relevant in the management of diarrhea and other gastrointestinal ailments occurring in farm animals.

Hence, the idea of studying the ENC in vitro effects on chicken intestinal contractility is the starting point for exploring its potential use in chicken farms. In our opinion, spontaneous motility represents an important target, as its modulation can bring benefits to animals. For this reason, we selected parameters related to the way in which the tissue lengthens/shortens (MCA and BSMA) and oscillates around its equilibrium position (SCV and PSD) in the different conditions (basal and in the presence of increasing concentrations of the extracts). SCV (standard deviation) is correlated to tone variability and provides a generic indication of the amplitude of the oscillations. Thanks to PSD analysis, we are able to extract the frequencies at which the tissue oscillates. The choice of identifying three frequency bands (LF, MF and HF) was made to simplify the analysis. Our hypothesis is that the low frequencies (LF) are related to the transit velocity of the bolus, while the high frequencies are related to spasm effects and pain. The meaning of the intermediate band is still subject to debate.

These aspects on contractility can also be highlighted in this species as a contribution to animal welfare.

We have therefore verified whether ENC has a direct action on spontaneous intestinal contractility to broaden the activity spectrum and support its use in feed supplementation.

In addition, Q, a different extract rich in condensed tannins, has been studied on the same parameters. The increase in low-frequency waves suggests a decrease in the bolus transit, which could lead to a reduction in fat absorption. Fats would thus be eliminated at a higher percentage.

The increase in contractility responsible for bolus mixing may favor the formation of mixed lipid micelles and the activity of pancreatic enzymes on proteins, lipids and carbohydrates, which would become more bioavailable for intestinal absorption. The effect of these extracts on the gallbladder is relevant. ENC and Q may increase bile excretion, which results in an increase in the ability to emulsify fats. The action on gallbladder contractility represents a parameter to monitor the correct metabolism of fats. A decrease in contractility, in fact, delays the bile flow towards the pancreatic duct, predisposing the chicken to biliary lithiasis. ENC has been shown to have no negative action on this parameter [6].

In the caecum, Q-induced increases in low frequencies may reduce the absorption of nutrients as the transit time reduces.

In the colon, the reduction in the low-frequency of contraction (starting from 0.5 mg/mL) by Q could be associated with a change in the transit velocity of the content, with a consequent increase in the absorption of electrolytes and an increase in the consistency of the stool. These beneficial effects are linked to the chemical composition of the phytocomplexes. In fact, it is well known that plants represent a valid starting point for the search for new chemotypes for the modulation of various parameters. Tannins can be used for the identification of new scaffolds as a starting point for the synthesis of molecules for the control of motility.

The beneficial effects are strictly related to phytocomplexes composition, which may include new chemotypes able to affect various biological targets. Tannin structures can inspire the design of new molecules for the control of intestinal motility. Furthermore, their multifaceted biological activity can be an added value in the treatment of complex pathologic disorders. The hydrolysable tannins can be considered as sort of codrugs, as they generate smaller bioactive molecules of interest such as gallic acid. Gallic acid exerts an anti-inflammatory action by inhibiting the production of pro-inflammatory cytokines [32] and interfering with the whole proinflammatory cascade [33]. As for condensed tannins, quercetin and its aglycone rutin have different biological effects, such as anti-inflammatory and antioxidants activities, resulting in cardiovascular protection. Similar properties have been reported analogously for natural benzodioxane-based lignans [34]. Furthermore, as xenobiotics, the constituents of phytocomplexes are metabolized in vivo, and the resultant metabolites can be, in turn, bioactive compounds [35].

For instance, ellagitannins and their metabolites, urolithines, have been reported to effectively affect several gastrointestinal functions [8,36,37,38]. Furthermore, isolated ellagitannins, such as corilagin [39], punicalagin [40] and their metabolites, such as Urolthin A [41] were shown to reduce inflammation in colitis. Therefore, these molecules may be a starting point for the synthesis of new compounds for pharmaceutical use. This is particularly important as hydrolysable tannins, characterized by a high number of chiral centers and a high molecular weight, require very complex syntheses, while their metabolites, such as ellagic acid and urolithins, generally have a more simple chemical structure [35].

Chestnut extract could have a better biological profile, in relation to its application in farm animals, since it influences smooth muscle gastrointestinal contractility in such a way as not to induce constipation in healthy animals, possibly providing the secondary metabolites endowed with many beneficial effects.

The identification of hydrolyzable tannin metabolites showing the same activities as the parent compounds may be an advantage for the development of new pharmaceutical products.

## 4. Materials and Methods

### 4.1. Chemicals

*Castanea sativa* Mill. wood (ENC) and *Schinopsis balansae* Engl. hardwood (Q) extracts used in this study were supplied by SilvaTeam, San Michele di Mondovì, Italy. All of them originate from a water extraction process.

### 4.2. Chemical Analyses

Tannin percentage was obtained by gravimetric analysis of vegetable tanning agents by using the filter Freiberg-Hide powder method. This is a standard method in the leather industry but is also used in other domains [42].

### 4.3. “In Vitro” Studies

#### 4.3.1. Animals

Fresh gastrointestinal tracts of healthy chicks Ross 308 (2.65–2.85 Kg) were obtained from a local slaughterhouse. The tissues (duodenum, caecum, ileum, proximal colon and gallbladder) required were immediately immersed in appropriate physiological salt solution (PSS) (see below). In laboratory, the tissues were rapidly set up under a suitable resting tension in 15 mL organ bath, containing PSS, consistently warmed (see below) and buffered to pH 7.4 by saturation with 95% O_2_–5% CO_2_ gas.

Caecum. Segment of caecum was removed and placed immediately in avian Ringer’s solution of the following composition (mM): NaCl, 110; KCl, 3.4; CaCl_2_, 2.6; MgSO_4_·7H_2_O, 0.88; NaH_2_PO_4_·H_2_O, 1.6; NaHCO_3_, 25; glucose, 15.0. The organ was cleaned and suspended under 1 g tension in organ bath maintained at 37 °C. The PSS was changed isometrically every 15 min for 60 min until complete stabilization.

Duodenum and Ileum. Duodenum and the terminal portion of ileum was cleaned, and segments 2 cm long were set up under 1 g tension in the longitudinal direction along the intestinal wall at 37 °C in organ baths containing a Tyrode solution of the following composition (mM): NaCl, 137.0; KCl, 5.4; CaCl_2_, 4.8; MgCl_2_, 1.1; NaH_2_PO_4_·H_2_O, 0.4; NaHCO_3_ 11.9; glucose 10.1. Tissues were allowed to equilibrate for at least 30 min during which time the bathing solution was changed every 10 min.

Gallbladder and proximal colon. The gallbladder and proximal colon were removed. Gallbladder was cleaned with Thyrode solution (see below) to remove bile residues and immediately set up for in vitro spontaneous contractility studies. The segments of about 1 cm of the proximal colon were cut at approximately 1 cm distal from the caecocolonic junction and cleaned by rinsing it with a Tyrode solution of the following composition (mM): NaCl, 118; KCl, 5.9; CaCl_2_, 2.5; MgSO_4_·7H_2_O, 1.2; NaH_2_PO_4_·H_2_O, 1.0; NaHCO_3_, 2.5; glucose, 10.0; and the mesenteric tissue was removed. The organs were suspended in organ baths containing gassed, warm PSS under a load of 1 g maintained at 37 °C. Tension changes in longitudinal muscle length were recorded. Tissues were allowed to equilibrate for at least 30 min during which time the bathing solution was changed every 10 min.

#### 4.3.2. Spontaneous Contraction

For the duodenum, caecum ileum, proximal colon and gallbladder, the tracing graphs of spontaneous contractions were continuously recorded with the LabChart Software (AD Instruments, Bella Vista, New South Wales, Australia). After the equilibration period (about 30 min to 45 min according to each tissue), cumulative-concentration curves (0.1, 0.5, 1.0, 5.0, 10.0 mg/mL) of tannin-based extracts were constructed. At the end of each single concentration, a 5-min stationary period was selected, and, for each interval, the following parameters were evaluated: mean contraction amplitude (MCA) calculated as the mean force value (g); the force contraction standard deviations, considered as an index of the spontaneous contraction variability (SCV); and basal spontaneous motor activity (BSMA), calculated as the percentage (%) variation of each mean force value (g) with respect to the control. The spontaneous contraction rates were investigated through a standard FFT analysis, and the evaluation of the absolute powers (inside a Power Spectral Density plot) of the following frequency bands of interest: low [0.0, 0.2] Hz, medium [0.2, 0.6] Hz and high [0.6, 1.0] Hz [43].

The mean tone, frequency, amplitude and duration of contractions of the distal proximal ileum, cecum, distal proximal colon and gallbladder were determined. All the calculations were performed in a post-processing phase with the Lab Chart Software. In order to avoid errors due to the presence of artifacts, the period of analysis was chosen by a skilled operator.

## 5. Conclusions

The action on gallbladder contractility represents a parameter to monitor the correct digestion and absorption of fat and bile acids. Similarly, the reduction of contractility in the ileum can suggest a better mixing of the ileal content and a better absorption of bile acids and nutrients, prompting the enterohepatic circulation, which is limited by increased transit time [44]. When the food reaches the ileum, proteins, fats and carbohydrates have been digested, leaving only some substances that can pass into the colon and be eliminated or enter the caeca. In chicken there are two, where bacteria help break down undigested food. Food is fermented for the production of organic acids, fatty acids and vitamins that chickens can absorb in addition to other nutrients; a reduction in tone allows a greater use of this mechanism. From the caeca, food moves to the large intestine, which absorbs water and dries out indigestible foods. The colon absorbs water, dries out indigestible foods and eliminates waste products.

The chemical analysis of the two extracts revealed that ENC mainly contains hydrolysable tannins, while Q mainly has condensed tannins. The two phytocomplexes show different effects on gastrointestinal smooth muscle contractility, with ENC providing a better activity profile than Q.

Both ENC and Q were shown to affect smooth muscle contractility in many gastrointestinal tracts with 2 main possible effects on the farm animals:A better digestion and absorption of nutrients.The formation of stools of better consistency.

Globally, it seems that ENC improves the nutrient adsorption with a better dose-dependent action than Q, making its use in chicken farms more manageable.

These data represent the starting point for further experiments aimed at evaluating the effects of Q and ENC in farm chickens. In particular, the results obtained describing the different effects on different intestinal tracts confirm how a combined use of ENC and Q can improve both the life quality of the animal and, at the same time, the quality of the meat through action on different targets, including contractility. Previous studies had yielded interesting in vivo results with the use of a mix of ENC and Q [21,45].

Tannins are a valid alternative to modulating the microbiota and modulating diarrhea. A reduction in the contractility of intestinal smooth muscle would promote absorption and cause an increase in fecal consistency.

This information, combined with the chemical composition of the extracts and information on the metabolism of these secondary metabolites, can be the starting point for the identification of new hits for the synthesis of chemical classes for pharmaceutical, animal and human use in the control of contractility.

The research proceeds with the evaluation of different combinations of ENC and Q to verify a potential synergistic effect.

## Figures and Tables

**Figure 1 molecules-28-00395-f001:**
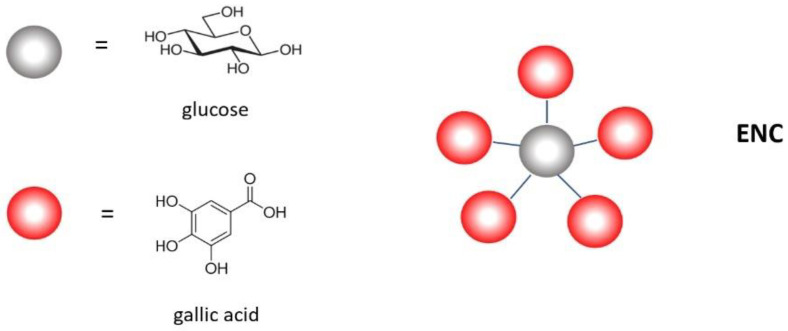
Most represented structures in ENC.

**Figure 2 molecules-28-00395-f002:**
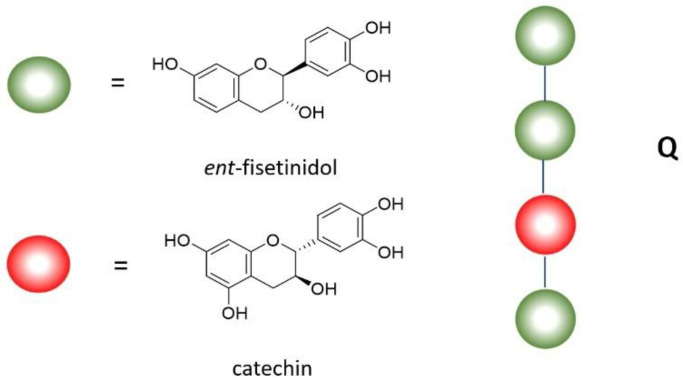
Most represented structures in Q.

**Figure 3 molecules-28-00395-f003:**
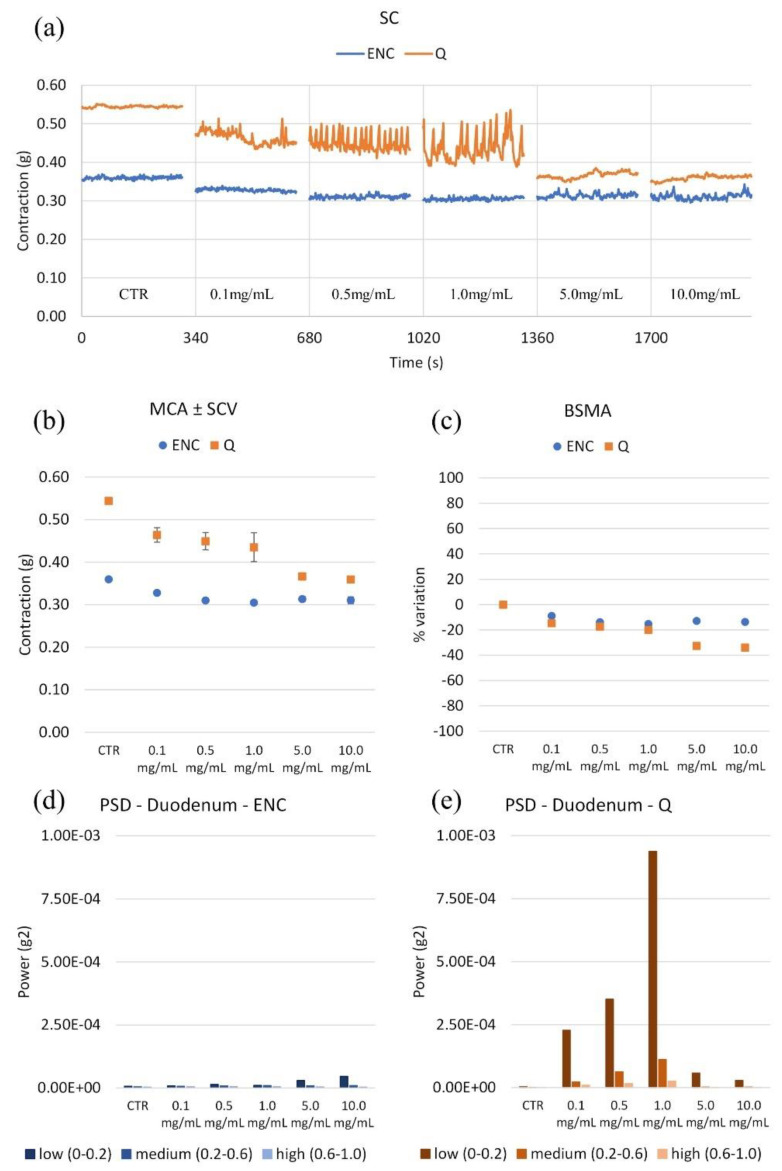
(**a**) Experimental original recording of the concentration–response curve of ENC (*Castanea sativa* Mill.) and Q (*Schinopsis balansae* Engl.) extract on duodenum spontaneous contractility (SC). (**b**) Mean contraction amplitude (MCA) with spontaneous contraction variability (SCV). (**c**) Basal Spontaneous Motor Activity (BSMA). Zero represents the basal tone, and each point is the percent variation from the baseline after cumulative addition of each dose; each value is the mean ± SEM; when the error bar is not shown, it is covered by the point. Power Spectral Density (PSD): (**d**) Duodenum with ENC; (**e**) Duodenum with Q.

**Figure 4 molecules-28-00395-f004:**
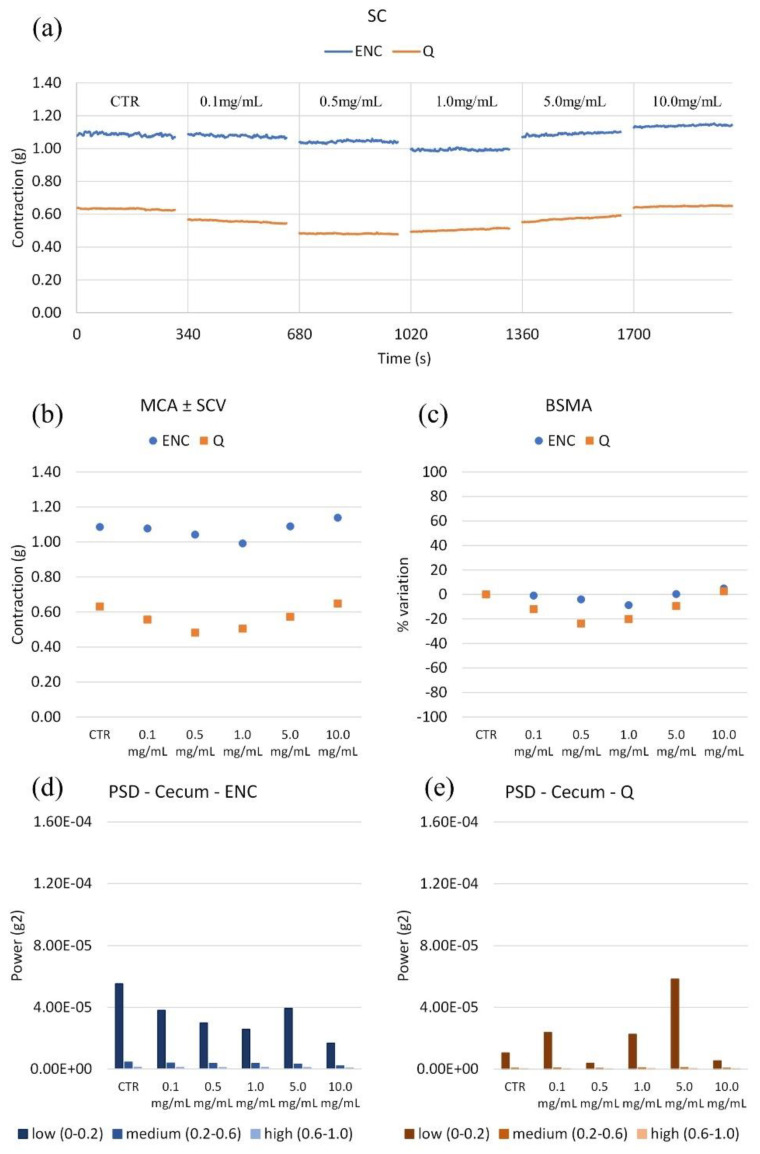
(**a**) Experimental original recording of the concentration–response curve of ENC (*Castanea sativa* Mill.) and Q (*Schinopsis balansae* Engl.) extract on caecum spontaneous contractility (SC). (**b**) Mean contraction amplitude (MCA) with spontaneous contraction variability (SCV). (**c**) Basal Spontaneous Motor Activity (BSMA). Zero represents the basal tone, and each point is the percent variation from the baseline after cumulative addition of each dose; each value is the mean ± SEM; when the error bar is not shown, it is covered by the point. (**c**) Power Spectral Density (PSD): (**d**) Caecum with ENC (**e**) Caecum with Q.

**Figure 5 molecules-28-00395-f005:**
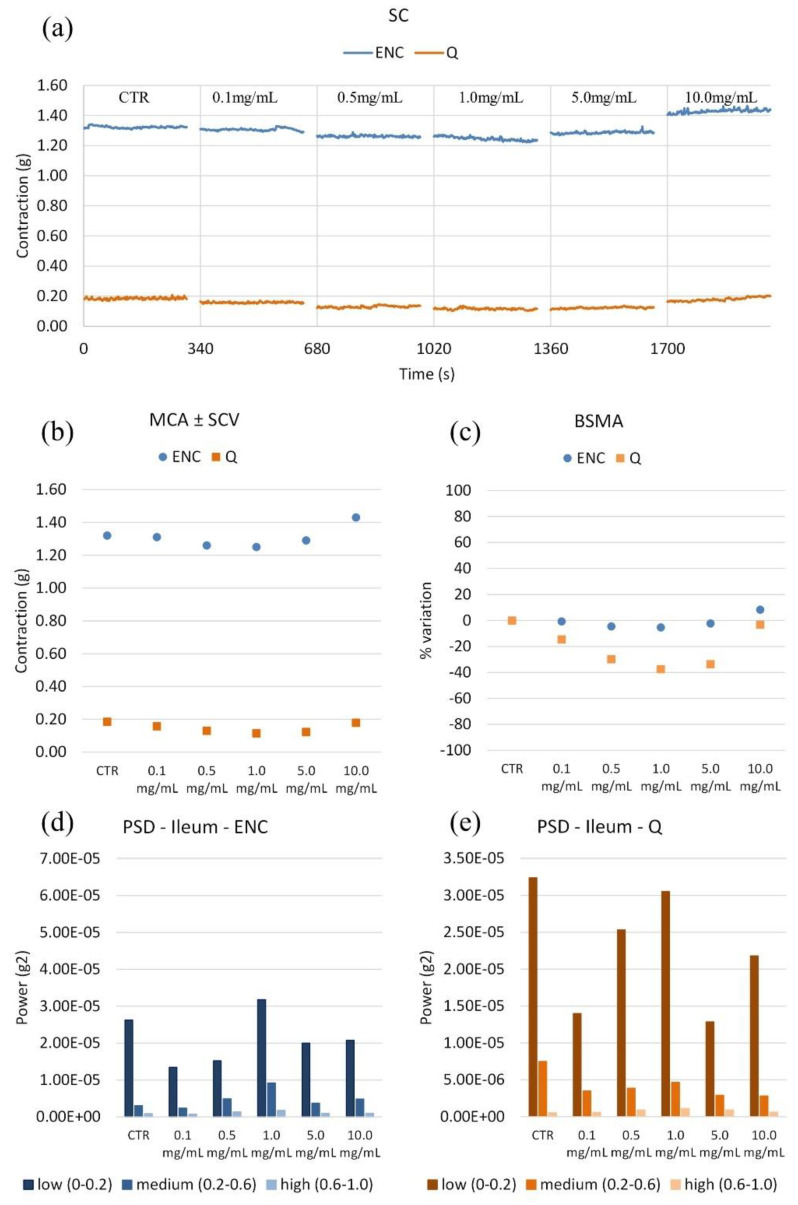
(**a**) Experimental original recording of the concentration–response curve of ENC (*Castanea sativa* Mill.) and Q (*Schinopsis balansae* Engl.) extract on ileum spontaneous contractility (SC). (**b**) Mean contraction amplitude (MCA) with spontaneous contraction variability (SCV). (**c**) Basal Spontaneous Motor Activity (BSMA). Zero represents the basal tone, and each point is the percent variation from the baseline after cumulative addition of each dose; each value is the mean ± SEM; when the error bar is not shown, it is covered by the point. (**c**) Power Spectral Density (PSD): (**d**) Ileum with ENC; (**e**) Ileum with Q.

**Figure 6 molecules-28-00395-f006:**
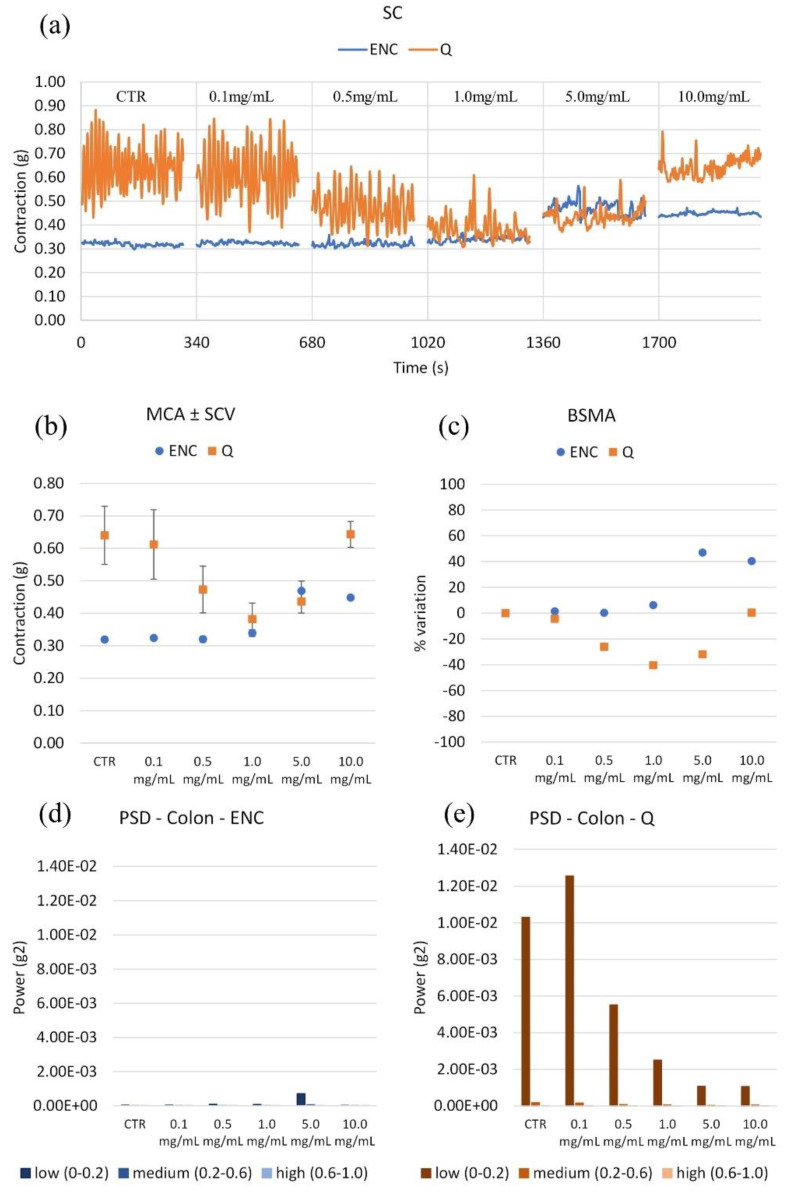
(**a**) Experimental original recording of the concentration–response curve of ENC (*Castanea sativa* Mill.) and Q (*Schinopsis balansae* Engl.) extract on colon spontaneous contractility (SC). (**b**) Mean contraction amplitude (MCA) with spontaneous contraction variability (SCV). (**c**) Basal Spontaneous Motor Activity (BSMA). Zero represents the basal tone, and each point is the percent variation from the baseline after cumulative addition of each dose; each value is the mean ± SEM; when the error bar is not shown, it is covered by the point. (**c**) Power Spectral Density (PSD): (**d**) Colon with ENC; (**e**) Colon with Q.

**Figure 7 molecules-28-00395-f007:**
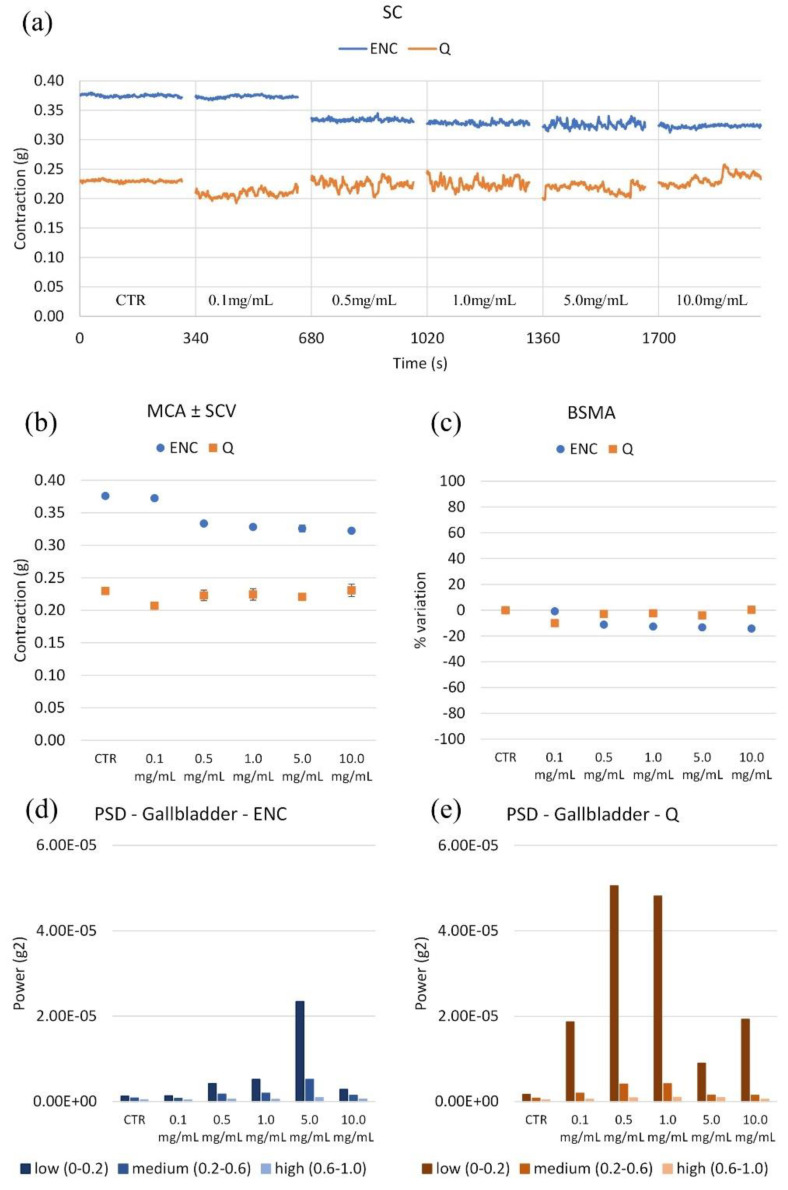
(**a**) Experimental original recording of the concentration–response curve of ENC (*Castanea sativa* Mill.) and Q (*Schinopsis balansae* Engl.) extract on gallbladder spontaneous contractility (SC). (**b**) Mean contraction amplitude (MCA) with spontaneous contraction variability (SCV). (**c**) Basal Spontaneous Motor Activity (BSMA). Zero represents the basal tone, and each point is the percent variation from the baseline after cumulative addition of each dose; each value is the mean ± SEM; when the error bar is not shown, it is covered by the point. (**c**) Power Spectral Density (PSD): (**d**) Gallbladder with ENC; (**e**) Gallbladder with Q.

**Table 1 molecules-28-00395-t001:** Composition of studied extracts (*w*/*w*).

Compound		Silvafeed ENC *Castanea sativa* Mill.	Silvafeed Q *Schinopsis balansae* Engl.
Tannins	Hydrolysable	76.70	
	Condensed		73.50
Non-tannins		17.30	14.00
Insoluble fraction		0.90	6.00
Crude fiber		<0.10	0.10
Ash		1.10	2.00
Moisture		5.10	6.50

**Table 2 molecules-28-00395-t002:** ENC and Q chemical compositions.

Tannins Composition	ENC	Q
Gallic acid	1–6%	1–2%
Ellagic acid	1%	−
Pentagalloyl glucose	3%	−
Castalin and vescalin	6.6%	−
Castalagin and vescalagin	~ 30%	−
Roburins	20–24%	−
Other major oligomers and glycosides	20–40%	−
Trigalloyl quinic acid	−	−
Tetragalloyl quinic acid	−	−
Pentagalloyl quinic acid	−	−
Esagalloyl quinic acid	−	−
Eptagalloyl quinic acid	−	−
Fisetinidin and robinetinidin dimers	−	9–11%
Fisetinidin and robinetinidin trimer	−	38–43%
Fisetinidin and robinetinidin tetramer	−	24–28%
Fisetinidin and robinetinidin pentamer	−	12–15%
Fisetinidin and robinetinidin esamers	−	9–12%

## Data Availability

Data is contained within the article, inside tables and graphs.

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
