# Peer review of "Tannins-Based Extracts: Effects on Gut Chicken Spontaneous Contractility"

_molecules, 2023, doi:10.3390/molecules28010395_

Round 1

Reviewer 1 Report

The authors presented a study on the effect of two natural tanni extracts on intestinal contractility in chickens.

The sections of the introduction, discussion and conclusion should be thoroughly revised, actually focusing on the problem, the objective and discussing the results appropriately.

The authors talk several times about the different biological effects of tannins throughout the text without any actual relationship between these and intestinal contractility. Furthermore, they repeatedly mention the prebiotic power of these extracts, linking it to the effect on intestinal contractility, without having any firm basis for this. It is unclear why the authors continually associate intestinal contractility with an antibiotic power of the products studied.

They also suggest that the association of the two extracts in question may be beneficial but without providing an argument as to why.

The authors claim to have carried out a chemical characterisation of the extracts, reporting data apparently only quoted from other works. Since the authors claim that the different chemical profile of the extracts results in a different function in intestinal contractility, I suggest adding some supporting data.

Furthermore, I suggest that the grammar be revised.

I suggest a major revision.

Title

I would suggest to reword the title in a more appropriate manner

Abstract

L16 “They are mixtures of polyphenolic compounds”, the term phytocomplex could fit better

L21 The results show that the chemical composition of the two phytocomplexes in-21 fluences the spontaneous intestinal contractility in different ways by regulating the tone and consequent progression of the food bolus.

How can you ascribe the function to the chemical composition?

L24 “an important target of the effects of tannins that contributes to 24 animal and indirectly to human health.”

How can you be so sure that this could have some effect on human health? Please reformulate the sentence.

Keywords: the first two are too long. Put , or ; between each keyword.

Introduction

The structure of the introduction is quite confused. The authors talk widely about the positive effect of chestnut extract (especially the antimicrobial) but barely mention or  discuss the choice to use the quebracho one. Moreover, the authors don’t go straight to the point.

L 37 Move the cite at the end of the sentence. There is a full stop in the middle of the sentence. Please put it in a correct form.

L60 in human diseased. Correct the sentence.

L67 Why are the authors constantly talking about the antibiotic properties of tannins when the aim of the study is to assess the effect on intestine contractility?

L74 And should be in small caps

Results

The authors report some data about the chemical composition, but it seems that these are just citation from other works. In this case the presented data are not results of the present study and should not be included in this section. Maybe this part it could be included in the introduction and cited in the M&M section.

Table 2  No caption is present.

Figures in the panels should be marked by letters.

Discussion

L209 Based on what do the authors say the products are the most promising? compared with what?

L215 Which positive effects?

Materials and methods

It is not clear whether the authors made a characterization of the extracts. Some results are reported but no material and method appear in the text.

Only the “Tannin percentage obtained by gravimetric analysis is reported. If the authors realized this assay I suggest to include a more detailed description of the method.

The title of the paragraph where the products are described should be entitled “Materials” or “Chemicals”

L 235 “may favore the”, please proofread grammar.

Conclusion

Conclusions should not contain results. Move table 3 to results.

Table 3. “Outlook .on tone modulation by extracts.” Please delete the full stop in the middle of the sentence

Based on what do you say ENC shows the most promising results? I would include a more extensive and detailed explanation in the discussion section.

Author Response

Title

I would suggest to reword the title in a more appropriate manner

Answer: The title was changed to “Effects of vegetal extracts towards chickens gastrointestinal contractility: a possible way to improve farm animals life quality and health”

Abstract

L16 “They are mixtures of polyphenolic compounds”, the term phytocomplex could fit better

Answer: L16 became L19.  The term phytocomplex replaced the sentence ““They are mixtures of polyphenolic compounds” as requested

L21 The results show that the chemical composition of the two phytocomplexes in-21 fluences the spontaneous intestinal contractility in different ways by regulating the tone and consequent progression of the food bolus.

How can you ascribe the function to the chemical composition?

Answer: L21 became L24.  Basing on previous results, we speculate the tannins are the main responsible for the effects on contractility, as we demonstrated ENC, rich in hydrolysable tannins, exert antimuscarinic, antihistamic, calcium antagonistic, activities in guinea pig ileum and colon. [Budriesi R, Ioan P, Micucci M, Micucci E, Limongelli V, Chiarini A. Stop Fitan: antispasmodic effect of natural extract of chestnut wood in guinea pig ileum and proximal colon smooth muscle. J Med Food. 2010 Oct;13(5):1104-10] Also condensed tannins were shown to modify gut contractility

L24 “an important target of the effects of tannins that contributes to 24 animal and indirectly to human health.”

How can you be so sure that this could have some effect on human health? Please reformulate the sentence.

The effects on contractility may lead to a reduction of liquid faeces that may result in a decrease of orofecal transmission of pathogenic microorganisms. Therefore, the studied phytocomplexes may be used in order to reduce antibiotics in farm animals, with a consequent reduction of selection of multidrug-resistant microbiological strains. Furthermore, these extracts are rich in antioxidant compounds that may provide a meat of better quality. Our final aim will be the identification of nutraceutical with long term salutogenic effects.

The sentence “In conclusion, we can hypothesize that tannins exert an interesting effect on contractility, an im-portant target of the effects of tannins that contributes to animal and indirectly to human health.” was changed to “The chemical analysis of the two extracts revealed that ENC mainly contains hydrolysable tannins while Q mainly has condensed tannins. The two phytocomplexes show different effects towards gastrointestinal smooth muscle contractility, with ENC providing a better activity profile than Q.”

 Keywords: the first two are too long. Put , or ; between each keyword.

We corrected as requested

Introduction

The structure of the introduction is quite confused. The authors talk widely about the positive effect of chestnut extract (especially the antimicrobial) but barely mention or discuss the choice to use the quebracho one. Moreover, the authors don’t go straight to the point.

Answer:

we explained the reasons why we used quebracho, adding the following sentence:

“Quebracho extract was used as it contains high amounts of condened tannins, endowed with many biologival beneficial effects. In addition, it was chosen in order to make a comparison between condensed and hydrolysable tannins, in order to understand which molecules may be further studied for applications in animals farming.” in lines 81-85

In order to better explain our aim , we added the sentence “

“…and were chemically characterized, in order to investigate their possible application in the field of industrial farms.” in lines 105-106

L 37 Move the cite at the end of the sentence. There is a full stop in the middle of the sentence. Please put it in a correct form.

Answer: L37 became L43.

The whole sentence was changed to:

“Therefore, the identification of nutaceuticals endowed with antimicrobial activi-ties and with other effects, such as antinflammatory, antioxidant and gut modulatory properties may lead to an improvement of general health status of animals and to a decrease of anitibiotics need, that may result in a reduction of antimicrobial resistance. [2]”

L60 in human diseased. Correct the sentence.

Answer: L60 became L71. “human diseased” was changed to “human diseases”

L67 Why are the authors constantly talking about the antibiotic properties of tannins when the aim of the study is to assess the effect on intestine contractility?

Answer: L 67 became L 85

We speculate that the effects on the reduction of pathogenic bacteria transmission may result from the formation of more solid stools and not from a direct antimicrobial effect, that, however, was reported for many molecules found in the extracts.

L74 And should be in small caps

Answer: L 74 became L 92

“And” was changed to “and”.

Results

The authors report some data about the chemical composition, but it seems that these are just citation from other works. In this case the presented data are not results of the present study and should not be included in this section. Maybe this part it could be included in the introduction and cited in the M&M section.

Answer: the results were reported because chemical composition was elucidated using the methods reported in the cited works and they were analysed also in our experiments. We added some pieces of information regarding the gravimetric analyses.

Table 2  No caption is present.

Answer: The following caption “Tannins and polyphenolic compounds concentration of ENC, and Q.” was written, as requested

Figures in the panels should be marked by letters.

Figures were replaced by the following figures, following referees’ statment

Figure 3

Figure 3. (a) Experimental original recording of the concentration-response curve of ENC (Castanea sativa Mill.), and Q (Schinopsis balansae Engl.) extract on duodenum spontaneous contractility (SC). (b) Mean contraction amplitude (MCA) with spontaneous contraction variability (SCV). Basal Spontaneous Motor Activity (BSMA). Zero represents the basal tone and each point is the percent variation from the baseline after cumulative addition of each dose; each value is the mean ± SEM; when the error bar is not shown, it is covered by the point. (c) Power Spectral Density (PSD).

Figure 4

Figure 4. (a) Experimental original recording of the concentration-response curve of ENC (Castanea sativa Mill.), and Q (Schinopsis balansae Engl.) extract on caecum spontaneous contractility (SC). (b) Mean contraction amplitude (MCA) with spontaneous contraction variability (SCV). Basal Spontaneous Motor Activity (BSMA). Zero represents the basal tone and each point is the percent variation from the baseline after cumulative addition of each dose; each value is the mean ± SEM; when the error bar is not shown, it is covered by the point. (c) Power Spectral Density (PSD).

Figure 5

Figure 5. (a) Experimental original recording of the concentration-response curve of ENC (Castanea sativa Mill.), and Q (Schinopsis balansae Engl.)extract on ileum spontaneous contractility (SC). (b) Mean contraction amplitude (MCA) with spontaneous contraction variability (SCV). Basal Spontaneous Motor Activity (BSMA). Zero represents the basal tone and each point is the percent variation from the baseline after cumulative addition of each dose; each value is the mean ± SEM; when the error bar is not shown, it is covered by the point. (c) Power Spectral Density (PSD).

Figure 6

Figure 6. (a) Experimental original recording of the concentration-response curve of ENC (Castanea sativa Mill.), and Q (Schinopsis balansae Engl.) extract on colon spontaneous contractility (SC). (b) Mean contraction amplitude (MCA) with spontaneous contraction variability (SCV). Basal Spontaneous Motor Activity (BSMA). Zero represents the basal tone and each point is the percent variation from the baseline after cumulative addition of each dose; each value is the mean ± SEM; when the error bar is not shown, it is covered by the point. (c) Power Spectral Density (PSD).

Figure 7

Figure 7. (a) Experimental original recording of the concentration-response curve of ENC (Castanea sativa Mill.), and Q (Schinopsis balansae Engl.) extract on gallbladder spontaneous contractility (SC). (b) Mean contraction amplitude (MCA) with spontaneous con-traction variability (SCV). Basal Spontaneous Motor Activity (BSMA). Zero represents the basal tone and each point is the percent variation from the baseline after cumulative addition of each dose; each value is the mean ± SEM; when the error bar is not shown, it is covered by the point. (c) Power Spectral Density (PSD).

All the captions were rewritten, as reported

Discussion

L209 Based on what do the authors say the products are the most promising? compared with what?

L 209 became L 248

The sentence “are probably the most promising” was changed to “may be promising tools”

L215 Which positive effects?

L 215 became L 254

The positive effects consist of the ability of tannins to affect smooth muscle contractility, to exert antioxidant, antinflammatory properties and affecting gut microbiota, favouring probiotics growth.

Materials and methods

It is not clear whether the authors made a characterization of the extracts. Some results are reported but no material and method appear in the text.

Only the “Tannin percentage” obtained by gravimetric analysis is reported. If the authors realized this assay I suggest to include  more detailed description of the method.

Answer: the tannin percentage was obtained by gravimetroic analysis of vegetable tanning compounds through the use of Filter-Freiberg Hide Powder Method and it was specified in the text.

The title of the paragraph where the products are described should be entitled “Materials” or “Chemicals”

Answer: Paragraph 4.1 was split in 2 subparagraphs

“4.1.1 Chemicals

Castanea sativa Mill. Wood (ENC), Schinopsis balansae Engl. hardwood (Q), extracts were used in this study was supplied by SilvaTeam, San Michele di Mondovì, Italy. All of them come from a water extraction process.

4.1.2 Chemical analyses Tannin percentage was obtained by gravimetric analysis of vegetable tanning agents by using the filter Freiberg-Hide powder method, this is a standard method for the leather industry, but also used in other domains [42].”

 L 235 “may favore the”, please proofread grammar.

Answer: L 235 became L 292

“may favore” was changed to “may favour”

Conclusion

Conclusions should not contain results. Move table 3 to results.

Answer: Table 3 was moved to results, as requested

Table 3. “Outlook .on tone modulation by extracts.” Please delete the full stop in the middle of the sentence

Answer: This table title was changed to “Outlook of ENC and Q effects in different gastrointestinal tracts.

Based on what do you say ENC shows the most promising results? I would include a more extensive and detailed explanation in the discussion section.

Answer: The whole discussion was revised as requested. In particular we explained that the 2 extracts were compared and they were shown to exert different effects, as shown in table 3, in different gastrointestinal segments. In particular, ENC was shown to have more promising results than Q as it reduces less than Q gastrointestinal contractility, providing effects that may result in a better absorption of nutrients and fats, without inhibiting gut contractions as strongly as Q. Indeed our aim was to evaluate a nutraceutical to be used in non-pathological conditions and not when diarrhoea has already set up.  This was reported in the Discussion section, through the following sentence:

“Chestnut extract could have a better biological profile, in relation to its application in farm animals, since it influences smooth muscle gastrointestinal contractility in such a way as not to induce constipation, in healthy animals, possibly providing the secondary metabolites endowed with many beneficial effects.” in lines 335 – 338.

Reviewer 2 Report

Dear authors, your work is very interesting, the comments I make below are for the purpose of improving your manuscript.

Lines 86-89. Integrate the objective and improve the writing.

Lines 92-93. Place in the materials and methods section.

Line 120. You forgot to put the table header. Please give it a title.

Lines 348-352. Improve the writing of the entire conclusion, there are parts that should go in the results section (i.e. table 3). Why put "in conclusion" in the conclusion?

As an important piece of information, it is necessary to conclude your work from the scope of the objectives, that is why it is important that they are also understood, therefore, this section is poorly presented and must be improved.

Author Response

Comments and Suggestions for Authors

Dear authors, your work is very interesting, the comments I make below are for the purpose of improving your manuscript.

Lines 86-89. Integrate the objective and improve the writing.

Answer: lines 86-89 became lines 104-110

The sentence “In addition, these extracts were studied on the gallbladder spontaneous contractility.

“Finally, the studied tannins are chemically characterized and may be a valid starting point for identifying new scaffolds to assist in the treatment of human diseases and to improve the health of livestock.” was changed to “In addition, these extracts were studied on the gallbladder spontaneous contrac-tility and were chemically characterized, in order to investigate their possible application in the field of industrial farms.”

Lines 92-93. Place in the materials and methods section.

Answer: lines 92-93 became lines 113-114

The sentence “Castanea sativa Mill. Wood (ENC), Schinopsis balansae Engl. hardwood (Q), extracts used in this study were supplied by SilvaTeam, San Michele di Mondovì, Italy.” was placed in materials and methods section, as requested.

Line 120. You forgot to put the table header. Please give it a title.

Answer: we added the following table title “Tannins and polyphenolic compounds concentration of ENC, and Q.”

Lines 348-352. Improve the writing of the entire conclusion, there are parts that should go in the results section (i.e. table 3). Why put "in conclusion" in the conclusion?

Answer: lines 348-352 became lines 417-421

The sentence “In conclusion, tannins exert their positive dietary effect in chickens not only for their antimicrobial action but also for effects on contractility as observed for humans. The experimental results of this preliminary study show that the most interesting are the hydrolyzable tannins of chestnut (ENC) followed by the condensed tannins of quebracho (Q) (Table 3).” was changed to “The chemical analysis of the two extracts revealed that ENC mainly contains hydrolysable tannins while Q mainly has condensed tannins. The two phytocomplexes show different effects towards gastrointestinal smooth muscle contractility, with ENC providing a better activity profile than Q.” following the referees’ suggestions.

As an important piece of information, it is necessary to conclude your work from the scope of the objectives, that is why it is important that they are also understood, therefore, this section is poorly presented and must be improved.

Answer: we thank the referee for this suggestion.

Concerning the  conclusions, in order to better explain the link between the effects on gastrointestinal contractility, the orofecal transmission and the nutritional status of healthy (and not diseased) chicken, the following part was added:

“Both ENC and Q were shown to affect smooth muscle contractility in many gastrointestinal tracts with 2 main possible effects on the farm animals:

  • a better digestion and absorption of nutrients
  • the formation of stools of better consistency

These data represent the starting point for further experiments aimed at evaluating the effects of Q and ENC in farm chickens”

Reviewer 3 Report

Molecules: Tannins based Extracts: Effects on Gut Chicken Contractility

The study is of scientifically intriguing. However, the manuscript might need some editing and improvement of language throughout the paper, and some points (not limited to the following ones) need to be concerned before it can be considered for publication.

1.     Title: Gut Chicken Contractility? Rewrite it.

2.     Abstract, and over all

Check the tense of statement, for example, L18, In this work we studied ….; L21, the results showed that….

The main results should be supplemented in the abstr.

L23-25, Please rephrase the conclusion. First, the conclusion should not be hypothesized. Second, the conclusion was hard to understand.

3. Introduction:

L37: do you want to express, which may be further investigated.

L63-66. Please rewrite it clearly.

L82-89. Merge them into one paragraph logically.

4.     Results:

Table 1. Please explain how to identify the hydrolysable and condensed tannin?

5.     Discussion

L79: replace “related to” with “such as”

L81-83: delete “Ndou et al (2015) also reported xylanases from different microbial origins differentially influenced the performance of growing pigs”.

Results

L201: many papers also describe their positive effects on farm…

L205, The Q also have ….

L237, change that to which

L244-245, rewrire it.

L254, change that to which

6.     Conclusion

Suggest to rewrite it plainly and simplistically.

L356-358, please rewrite it.

Author Response

Comments and Suggestions for Authors

Molecules: Tannins based Extracts: Effects on Gut Chicken Contractility

The study is of scientifically intriguing. However, the manuscript might need some editing and improvement of language throughout the paper, and some points (not limited to the following ones) need to be concerned before it can be considered for publication.

  1. Title: Gut Chicken Contractility? Rewrite it.

The title “Tannins based Extracts: Effects on Gut Chicken Contractility” was changed to “Effects of vegetal extracts towards chickens gastrointestinal contractility: a possible way to improve farm animals life quality and health”

  1. Abstract, and over all

Check the tense of statement, for example,

L18, In this work we studied ….; L21, the results showed that….

Answer:

Answer:

The sentence “They are mixtures of polyphenolic compounds obtained from natural matrices with multiple health properties used both for human health and as feed supplement in chicken farms.” was changed to “These phytocomplexes obtained from natural matrices with multiple health properties may be used both for human health and as feed supplement in chicken farms.”

L18 became L21. “study” was changed to “studied”

L21 became L24L. “show” was changed to “showed”

The main results should be supplemented in the abstr.

Answer: In order to summarize the results, as suggested, the sentence “In conclusion, we can hypothesize that tannins exert an interesting effect on contractility, an im-portant target of the effects of tannins that contributes to animal and indirectly to human health.” was replaced by “The chemical analysis of the two extracts revealed that ENC mainly contains hydrolysable tannins while Q mainly has condensed tannins. The two phytocomplexes show different effects towards gastrointestinal smooth muscle contractility, with ENC providing a better activity profile than Q.”

L23-25, Please rephrase the conclusion. First, the conclusion should not be hypothesized. Second, the conclusion was hard to understand.

Answer: we thank the referee for this suggestion. The sentence “In conclusion, we can hypothesize that tannins exert an interesting effect on contractility, an im-portant target of the effects of tannins that contributes to animal and indirectly to human health.” was replaced by “The chemical analysis of the two extracts revealed that ENC mainly contains hydrolysable tannins while Q mainly has condensed tannins. The two phytocomplexes show different effects towards gastrointestinal smooth muscle contractility, with ENC providing a better activity profile than Q.”

  1. Introduction:

L37: do you want to express, which may be further investigated.

Answer: L37 became L47

The whole paragraph in lines 36-38 was rewritten ad replaced by the following sentence:

“Therefore, the identification of nutaceuticals endowed with antimicrobial ac-tivi-ties and with other effects, such as antinflammatory, antioxidant and gut modu-latory properties may lead to an improvement of general health status of animals and to a decrease of anitibiotics need, that may result in a reduction of antimicrobial resistance.”

The sentence “May be futher investigated”   had not been deleted from a first draft

L63-66. Please rewrite it clearly.

Answer: L63-66 became L 77-80

The sentence “Moreover, this extract combines cardiovascular protective effects [7], also in animal models treated with a high fat diet [8] highlighting interesting anti-inflammatory ef-fectsoccurring systematically and in the intestinal district [9].” was changed to “Other activities of this phytocomplex that may result in a quality improvement of animals health status are the antinflammatory and antioxidant effects. In particular, this phytocomplex was shown to exert cardiovascular protective effects in healthy and obese animals. [7-9]”

L82-89. Merge them into one paragraph logically.

Answer: L82-89 became L97-106

This paragraph was changed to “The aim of this work was to study the in vitro effects of an hydrolysable tannins rich extract from Castanea sativa Mill. (Silvafeed ENC) and a condensed tannins rich extract from Schinopsis balansae Engl. (Silvafeed Q) on the intestinal basal contractility of healthy chickens. In addition, these extracts were studied on the gallbladder spontaneous contractility and were chemically characterized, in order to investigate their possible application in the field of industrial farms.”

  1. Results:

Table 1. Please explain how to identify the hydrolysable and condensed tannin?

Answer: the table title was changed to “Table 1. Gravimetric data of the two extracts ENC and Q expressed as percentage dry weight” in order to specify the analytical methos used for these analyses.

  1. Discussion

L79: replace “related to” with “such as”

Answer: we corrected as suggested

L81-83: delete “Ndou et al (2015) also reported xylanases from different microbial origins differentially influenced the performance of growing pigs”.

Answer:we deleted the sentence, as requested

Results

L201: many papers also describe their positive effects on farm…

Answer: L201 became L 257. The sentence “ are currently used in humans but many papers describe their”  was replaced by “were shown to exert”

L205, The Q also have ….

Answer: L205 became L 262. “have” was changed in “has”

L237, change that to which

Answer: L237 became L 294. “that” was changed to “which”.

L244-245, rewrire it.

Answer: L244 became L 301.

The sentence “In the caecum, Q induced increase in the low frequencies may promote the ab-sorption of nutrients as in this tract favors the transit of the content and reabsorption of water and electrolytes occurs.” was changed to “In the caecum, Q induced-increase in low frequencies may promote the absorption of nutrients the transit of the content and water and electrolytes reabsorption.”

L254, change that to which

Answer: L254 became L 313.

“that” was replaced by “which”

  1. Conclusion

Suggest to rewrite it plainly and simplistically.

Answer: conclusions were revised, as suggested, with the following modifications:

The action on gallbladder contractility represents a parameter to monitor the correct digestion and absorption of fat and bile acids. Similarly, the reduction of contractility in the ileum can suggest a better mixing of the ileal content and a better absorption of bile acids and nutrients, prompting also the enterohepatic circulation, that is limited by increased transit time [44]. When the food reaches the ileum, proteins, fats and carbohydrates have been digested, leaving only some substances that can pass into the colon and be eliminated or enter the caeca, that in chicken are two, where bacteria help break down undigested food. Food is fermented for the production of organic acids, fatty acids and vitamins that chickens they can absorb in addition to other nutrients: a reduction in tone allows a greater use of this mechanism. From the caeca, food moves to the large intestine, which absorbs water and dries out indigestible foods. The colon absorbs water, dry out indigestible foods and eliminate waste products.

The chemical analysis of the two extracts revealed that ENC mainly contains hydrolysable tannins while Q mainly has condensed tannins. The two phytocomplexes show different effects towards gastrointestinal smooth muscle contractility, with ENC providing a better activity profile than Q.

In conclusion, tannins exert their positive dietary effect in chickens not only for their antimicrobial action but also for effects on contractility as observed for humans. The experimental results of this preliminary study show that the most interesting are the hydrolyzable tannins of chestnut (ENC) followed by the condensed tannins of quebracho (Q) (Table 3).

Table 3. Outlook .on tone modulation by extracts.

Extract

Tannins

duodenum

caecum

ileum

colon

gallbladder

ENC

Hydrolysable

(↔)

(↑)

(↓/↔)

Q

Condensed

(↓↓)

(↓)

↓↓

(↔)

Legend: ↑, increase; ↓, decrease; ↓↓, remarkable decrease; ↔, constant.

Both ENC and Q were shown to affect smooth muscle contractility in many gastrointestinal tracts with 2 main possible effects on the farm animals:

  • a better digestion and absorption of nutrients
  • the formation of stools of better consistency

These data represent the starting point for further experiments aimed at evaluat-ing the effects of Q and ENC in farm chicken.

In particular, the results obtained describing different effects on different intestinal tracts confirm how a combined use of ENC and Q can improve both the life quality of the animal and at the same time the quality of the meat through action on different targets including contractility. Previous studies had yielded interesting in vivo results with the use of a mix of ENC and Q [21,45].

Tannins are a valid alternative to modulating the microbiota and modulating diarrhea. A reduction in the contractility of intestinal smooth muscle would promote absorption and cause an increase in fecal consistency.

This information, combined with the chemical composition of the extracts and information on the metabolism of these secondary metabolites, can be the starting point for the identification of new hits for the synthesis of chemical classes for pharmaceutical, animal and human use, for the control of contractility.

The research proceeds with the evaluation of different combinations of ENC and Q to verify a potential synergistic effect.

Reviewer 4 Report

Congratulations on the work, I would like to indicate just some little modifications to be applied:

+ Please check some strapped words in the manuscript and an English revision is recommended, particularly for figures and tables titles.

for instance: in Line 70,  I do not understand "comatch"

+ please redescribe well the title of table 1, with an indication of the employed unit.

+ Line 120, please eliminate the sentence which is surely originated from the template.

+ Line 353, please apply an adequate title for the table. Moreover, it appears that this section was moved down to the conclusion because, where we are not expecting tables and such discussion. 

Author Response

Comments and Suggestions for Authors

Congratulations on the work, I would like to indicate just some little modifications to be applied:

+ Please check some strapped words in the manuscript and an English revision is recommended, particularly for figures and tables titles.

Answer: we thank the referee for this suggestion. The following titles were added to the manuscript:

In table 1, the following caption “Components percentage obtained by gravimetric analysis” was added

The following table 2 title was added: “Chemical composition of ENC and Q extracts”

The entire manuscript has been proofread, through a careful analysis of the language and grammar. There are changes throughout the manuscript, according to the referee’s suggestions

for instance: in Line 70,  I do not understand "comatch"

Answer: L70 became L 88. “comatch” was replaced by “fit with”

+ please redescribe well the title of table 1, with an indication of the employed unit.

Answer: The title “Composition of studied extracts (w/w).” was changed to “Gravimetric data of the two extracts ENC and Q expressed as percentage dry weight”

+ Line 120, please eliminate the sentence which is surely originated from the template.

Answer: L 120 became L142

The sentence “Table 2. This is a table. Tables should be placed in the main text near to the first time they are cited.” was deleted, as requested.

+ Line 353, please apply an adequate title for the table. Moreover, it appears that this section was moved down to the conclusion because, where we are not expecting tables and such discussion.

Answer: L353 became L422

The title “Outlook on tone modulation by extracts.” was changed to “Outlook of ENC and Q effects in different gastrointestinal tracts” and table 3 was moved to results.

Round 2

Reviewer 1 Report

The authors partially modified the writing. However, this should be further improved for it to be suitable for publication.

Title: 

Since tannins is one of the key words in the paper, it should appear in the title.

Abstract:

L 21 It cannot be grasped that tannin extracts can be useful for human health when the work is centered totally on chickens. It is off topic, so it should be deleted.

L27 No abbreviations that are not conventional should appear in the abstract.

Keywords:

I suggest a more appropriate way to refer to extracts such as : Tannins; Quebracho; Chestnut.

A “;” should be put instead of “,” between the various gut segments.

Introduction

L43 Correct the word “nutaceuticals”

L63 Nowadays, aren't tannins derived from quebracho also used for feed?

L81 Correct the word “condened”

L96 The sentence about the application in humans is taken out of context. Please remove it.

L100 “Tannins” should not have the final s, as in line 102

L104 Review the grammar, there is a period in the middle of the sentence.

Results

L116 add , before respectively

L118- L143 The gravimetric method does not provide detailed information on the chemical composition of the two extracts. In fact, this method allows only the results shown in Table 1 to be obtained. As suggested in the previous review, results from other previous studies cannot be reported in the results section. This is not a review. This part should therefore be removed and used in more appropriate sections such as the introduction to describe the extracts used or used as support for discussion. This observation is about the text, Table 2 and Figures 1 and 2.

L134 The abbreviation has been inserted before. Remove (Q)

L154 Table 3 is not mentioned anywhere in the text.

In addition, the is not stated what the arrows indicate. What parameter or concentration do they refer to? In comparison to what? What do the parentheses indicate? Please specify in the text both in the results section and in the discussion.

L159 Explain in comparison to what there is decrease in spontaneous contractility. 

Specify in a clearer sentence this aspect “Q >> ENC”

Figure 3 to Figure 7. In the previous revision, letters were requested to be added to better read the graph panel, and the authors added only 3 letters. Please add a letter er each panel image (5 in total) next to the title of each.

Discussion

After a long introduction (L249-L278), only a very small part of the text is devoted to a discussion of the results. I suggest reviewing this part and clarifying some concepts that are mentioned in the results. For example, what is the significance and influence of shrinkage variability and all the other evaluated parameters? Why were they chosen? It would be ideal to more deeply discuss the changes induced on each by the extracts studied.

L262 Chestnut extract has also presented numerous beneficial properties for humans. I suggest writing a summary sentence that includes the two extracts.

L285 Too many repetition of “in addition”.

L285 The authors wrote “In addition, the effects of the two extracts were compared in order to identify a relationship between the chemical composition and the biological activity.”

Nowhere in the text is the role of the chemical composition of the extracts on the different influence on intestinal contractility clarified. To do so, it would be necessary to show that some molecules actually belonging to the respective extracts perform this function. The authors report only a gravimetric method that is used to detect the presence of hydrolysable and non-hydrolysable tannins, information that is already known.

Please better clarify this point.

L288 Which extract are you talking about?

L289 Correct “effct”.

L308 How is this demonstrated?

L307-L302 Please rewrite this concept.

Discussion L313 to 334 is beyond the scope of the topic; one could summarize this part and devote more space to the discussion of the results obtained that are seen only superficially.

L335 to L338. Remove quotation marks in paragraph. As in the previous review, it does not appear clear how in the end it turns out that chestnut extract turns out to be better than quebracho extract. I suggest adding a final comparative summary explanation. In the paragraph on L305 it appears that quebracho has a positive effect on electrolyte absorption and stool consistency.

Materials and Methods. 

The numbering should be :

4.1 Chemicals

4.2 Chemical analyses

4.3 In vitro studies

Conclusions

The text in the conclusion could be useful material for discussion. 

The conclusion section should summarize only the main findings, possible limitations and/or future implications. No other work should be cited in the conclusion section. 

Author Response

Comments and Suggestions for Authors

The authors partially modified the writing. However, this should be further improved for it to be suitable for publication.

Title:

Since “tannins” is one of the key words in the paper, it should appear in the title.

We agree with the referee. We apologize for the incorrect edit made. We changed the title as required.

Abstract:

L 21 It cannot be grasped that tannin extracts can be useful for human health when the work is centered totally on chickens. It is off topic, so it should be deleted.

The sentence has been removed.

L27 No abbreviations that are not conventional should appear in the abstract.

Abbreviations in the abstract have been removed.

Keywords:

I suggest a more appropriate way to refer to extracts such as : Tannins; Quebracho; Chestnut.

A “;” should be put instead of “,” between the various gut segments.

Keywords have been changed as required.

Introduction

L43 Correct the word “nutaceuticals”

The word has been corrected.

L63 Nowadays, aren't tannins derived from quebracho also used for feed?

The sentence has been modified as required.

L81 Correct the word “condened”

The word has been corrected.

L96 The sentence about the application in humans is taken out of context. Please remove it.

The sentence has been removed.

L100 “Tannins” should not have the final s, as in line 102

The word has been corrected.

L104 Review the grammar, there is a period in the middle of the sentence.

 The grammar has been corrected.

Results

L116 add , before respectively

The comma has been inserted.

L118- L143 The gravimetric method does not provide detailed information on the chemical composition of the two extracts. In fact, this method allows only the results shown in Table 1 to be obtained. As suggested in the previous review, results from other previous studies cannot be reported in the results section. This is not a review. This part should therefore be removed and used in more appropriate sections such as the introduction to describe the extracts used or used as support for discussion. This observation is about the text, Table 2 and Figures 1 and 2.

We agree with the referee but, for editorial reasons, we prefer to include the chemical composition of the extracts in this point as it is not possible to put this info in the introduction and the materials and methods section is after the results and therefore not functional to understanding the results and the discussion. However, all bibliographic references have been added and it is clear that the characterization of the extracts has been done previously.

L134 The abbreviation has been inserted before. Remove (Q)

The sentence has been modified.

L154 Table 3 is not mentioned anywhere in the text.

In addition, the is not stated what the arrows indicate. What parameter or concentration do they refer to? In comparison to what? What do the parentheses indicate? Please specify in the text both in the results section and in the discussion.

We agree with the referee that table 3 is not easy to understand. For this reason we decided to remove it and insert more details in the text aimed at better understanding the effects of the two extracts.

L159 Explain in comparison to what there is decrease in spontaneous contractility.

Specify in a clearer sentence this aspect “Q >> ENC”

The symbol >> has been removed and replaced with a sentence.

Figure 3 to Figure 7. In the previous revision, letters were requested to be added to better read the graph panel, and the authors added only 3 letters. Please add a letter er each panel image (5 in total) next to the title of each.

The required letters in the figure panels have been added.

Discussion

After a long introduction (L249-L278), only a very small part of the text is devoted to a discussion of the results. I suggest reviewing this part and clarifying some concepts that are mentioned in the results. For example, what is the significance and influence of shrinkage variability and all the other evaluated parameters? Why were they chosen? It would be ideal to more deeply discuss the changes induced on each by the extracts studied.

A sentence was added to clarify the meaning of the evaluated paramters.

L262 Chestnut extract has also presented numerous beneficial properties for humans. I suggest writing a summary sentence that includes the two extracts.

The sentence has been changed.

L285 Too many repetition of “in addition”.

The repetition “In addition” has been deleted.

L285 The authors wrote “In addition, the effects of the two extracts were compared in order to identify a relationship between the chemical composition and the biological activity.”

Nowhere in the text is the role of the chemical composition of the extracts on the different influence on intestinal contractility clarified. To do so, it would be necessary to show that some molecules actually belonging to the respective extracts perform this function. The authors report only a gravimetric method that is used to detect the presence of hydrolysable and non-hydrolysable tannins, information that is already known.

Please better clarify this point.

The sentence previously entered was actually incorrect. The effects according to the individual components have not been evaluated: we only wanted to highlight that the different composition of the extracts is related to the effects. For this reason, we have eliminated the phrase from the text.

L288 Which extract are you talking about?

The sentence refers to Q and so it was inserted in the previous paragraph.

L289 Correct “effct”.

The word has been corrected.

L308 How is this demonstrated?

The effects given by the extracts whose chemical composition is known represent the starting point for the identification of the single components and their chemical characterization aimed at a pharmaceutical use. The extract, on the other hand, is a nutraceutical use. It is therefore an observation linked to the type of journal that does not require further demonstrations in this article.

L307-L302 Please rewrite this concept.

The sentence was modified specifying the tissue of interest (colon) at the beginning.

Discussion L313 to 334 is beyond the scope of the topic; one could summarize this part and devote more space to the discussion of the results obtained that are seen only superficially.

This paragraph is necessary due to the journal’s scope and cannot be eliminated. Comments to the results have been directly inserted in the result section. In the discussion we resume what it has already discussed previously. Some comments represent an hypothesis of the authors already presented and accepted in other papers.

L335 to L338. Remove quotation marks in paragraph.

Done.

As in the previous review, it does not appear clear how in the end it turns out that chestnut extract turns out to be better than quebracho extract. I suggest adding a final comparative summary explanation. In the paragraph on L305 it appears that quebracho has a positive effect on electrolyte absorption and stool consistency.

A summary of the effects of ENC and Q has been inserted in the “Conclusions” section. However it is important to underline that both ENC and Q could have positive effects, it is not possible to state that ENC is good and Q is bad. After considering all the tissues, We can only suggest that ENC can be more useful and effective than Q, since its effects are more controllable (dose-dependent).

Materials and Methods.

The numbering should be :

4.1 Chemicals

4.2 Chemical analyses

4.3 In vitro studies

Paragraph numbering has been changed as required.

Conclusions

The text in the conclusion could be useful material for discussion.

The conclusion section should summarize only the main findings, possible limitations and/or future implications. No other work should be cited in the conclusion section.

A sentence was added to better clarify our findings.

Reviewer 3 Report

The author has made efforts to revise the manuscript according to the most of comments. Generally, the correction makes the manuscript better. However, there are few details, especially, some wording and sentence structure, that still need revision before it could be proceeding for publication.

Author Response

The author has made efforts to revise the manuscript according to the most of comments. Generally, the correction makes the manuscript better. However, there are few details, especially, some wording and sentence structure, that still need revision before it could be proceeding for publication.

The manuscript was revised and spelling errors corrected.